# Clover Dry Matter Predictor Based on Semantic Segmentation Network and Random Forest

Yin Ji [1] , Jiandong Fang [1,2,*] and Yudong Zhao [2]

1   College of Information Engineering, Inner Mongolia University of Technology, Hohhot 010080, China; jiyin345@gmail.com
2   Inner Mongolia Key Laboratory of Perceptual Technology and Intelligent Systems, Hohhot 010080, China
*   Correspondence: fangjd@imut.edu.cn

**Abstract:** As a key animal feed source, the dry matter content of clover is widely regarded as an important indicator of its nutritional value and quality. The primary aim of this study is to introduce a methodology for forecasting clover dry matter content utilizing a semantic segmentation network. This approach involves constructing a predictive model based on visual image information to analyze the dry matter content within clover. Given the complex features embedded in clover images and the difficulty of obtaining labeled data, it becomes challenging to analyze the dry matter content directly from the images. In order to address this issue, a method for predicting dry matter in clover based on semantic segmentation network is proposed. The method uses the improved DeepLabv3+ network as the backbone of feature extraction, and integrates the SE (Squeeze-and-Excitation) attention mechanism into the ASPP (Atrous Spatial Pyramid Pooling) module to enhance the semantic segmentation performance, in order to realize the efficient extraction of the features of clover images; on this basis, a regression model based on the Random Forest (RF) method is constructed to realize the prediction of dry matter in clover. Extensive experiments conducted by applying the trained model to the dry matter prediction dataset evaluated the good predictor performance and showed that the number of each pixel level after semantic segmentation improved the performance of semantic segmentation by 18.5% compared to the baseline, and there was a great improvement in the collinearity of dry matter prediction.

**Keywords:** clover dry matter prediction; DeepLabv3+; Squeeze-and-Excitation attention mechanism; random forest; deep learning

## 1. Introduction

Clover is one of the major sources of feed for livestock. It provides an abundance of nutrients, including carbohydrates, proteins, vitamins and minerals, to meet the nutritional needs of livestock for growth, development and production. The right clover feed can improve livestock performance and health. Growing high quality clover can provide a stable feed supply, reduce feed costs and increase the benefits of livestock farming. Real-time prediction of the dry matter content of clover can effectively improve the supervision of clover quality in the field. As an important source of animal feed, the dry matter content of clover is a key indicator of its nutritional value and quality. By accurately predicting the dry matter content of clover, it can help farmers and livestock farmers to determine the appropriate feed supply, which is crucial for the development of the livestock industry [1–3].

The traditional method of dry matter content determination involves cumbersome operations in several stages, and the method that is usually used requires a series of operations such as clover planting, cultivation, harvesting, sun-drying and weighing [4,5], which is a long process and often requires clover to be weighed throughout the entire growth cycle before the final clover content level can be obtained, which makes it difficult

to reflect the dry matter content level of clover in a timely manner during the growth process, and is difficult to reflect the dry matter content level of clover during the growth process at a time–cost. In terms of time and cost, it is difficult to respond to the dry matter content level of clover in the growth process in a timely manner, and due to the destructive operation of harvesting clover, it is impossible to continuously obtain the dry matter content of clover in the same area. Consequently, this constraint impedes the seamless implementation of real-time monitoring and control over the developmental trajectories of clover.

As an information carrier, an image contains rich visual and semantic information. By analyzing clover images, we can obtain the coverage of clover and the sparse density of clover growth [6,7], in addition to the use of clover images which can be used to classify various types of clover [8,9]; based on the image of clover, we also hope to obtain more information from the image, especially the depth of the dry matter content information contained in the image of clover.

There have also been many approaches to use images to predict dry matter content in clover, for example, Skovsen et al. [10] achieved pixel-by-pixel classification of clover, grass and weeds by training a full convolutional neural network, and then constructed a linear regression model based on the results of the semantic segmentation to predict the dry matter content, while Mortensen et al. [11] classified clover and grasses by color features, using edge detection and morphology to segment clovers and grasses and thus predict total dry matter, while Bretas et al. [12] combined satellite images and meteorological data to predict above-ground biomass and dry matter content. Spectral images have also been used for prediction; Murphy et al. [13] used near-infrared spectra to predict the mass and dry matter content of fresh grass, and Sun et al. [14] combined spectra and height to estimate biomass and nutritive value. Albert et al. [15] trained a perceptual segmentation network with synthesized imagery, and then trained a regression CNN with the generated labels to predict hay clover biomass. Hjelkrem et al. [16] developed a process-based model "NORNE" to predict the dry matter yield of grassland using remote sensing information from a drone, while Tosar et al. [17] used a near-infrared (NIR) imaging system to identify red clover and estimate the weight.

Neural networks have achieved remarkable success in domains such as computer vision and related fields [18–20]. The shortcoming of using them for the task of clover dry matter prediction is that labeling the dry matter content of each pixel, attributable to the possible presence of noise, occlusion and illumination variations in the image, generates a random error that affects the prediction performance. This study is carried out through the amalgamation of the potent feature extraction prowess inherent in semantic segmentation networks [21–24] and the excellent regression performance of RF, and by borrowing the concept of transfer learning [25–27]. In this research endeavor, we introduce a predictive methodology that hinges on the synergy between semantic segmentation networks and RF algorithms, which is able to efficiently analyze the dry matter prediction of clover images at the pixel level.

DeepLabv3+, an advanced semantic segmentation network with excellent pixel-level feature extraction and image segmentation capabilities, is first used. By associating each pixel in the image with a specific semantic label, it was possible to accurately distinguish clover, background and other elements. This provides a strong basis for subsequent dry matter content prediction.

Further, RF regression is introduced as a complement to further dry matter prediction in clover images. RF is an integrated learning method, which consists of multiple decision trees and can effectively handle complex nonlinear relationships. By combining the feature representation of the semantic segmentation network and the regression capability of RF, the dry matter content of each pixel in clover images can be predicted more accurately.

The main contributions and innovations of this paper are as follows:

- It obtains the rich feature representation of clover images through semantic segmentation network, and then combines it with RF regression to construct a dry matter

prediction model of clover images to realize the function of predictive analysis of dry matter content.

- It uses the DeepLabv3+ network with MobileNetv2 as the backbone as the feature extraction network and uses the SE attention mechanism to improve the ASPP, which, compared with the FCN-8s model used by the producer of the open-source dataset, has an improvement of up to 18% in the mIoU collinearity for the semantic segmentation task of the GressClovers dataset. has an improvement of up to 18.5%.
- It obtains the pixel-level features of the species in the image through semantic segmentation to understand the semantic information of the various classes in the image, which is used to obtain the deep information linking the distribution of the species in the image to the dry matter by constructing a RF regression model.
- Provides a new, low-cost and efficient solution for the prediction of dry matter in clover.

The remainder of this paper is organized into three sections. The second section will detail the model used in this paper while explaining the methodology and content within the model. In Section 3, we delve into the specifics of the experimental design. Finally, Section 4 presents a comprehensive discussion of the experimental results and offers insights into the future prospects of the application.

## 2. Methods

### 2.1. Model Overall Architecture

Figure 1 demonstrates the overall framework of clover dry matter prediction; the DeepLabv3+ network uses the Atrous Spatial Pyramid Pooling, adept at capturing semantic intricacies within images. Features extracted within the Encoder component undergoes a bifurcation into high-level and low-level variants, providing a versatile means of feature extraction across diverse scales. The pivotal inquiry rests in the capacity of the DeepLabv3+ semantic segmentation model to aptly convey its image features into the dry matter prediction model. In this study's designed dry matter prediction model, the DeepLabv3+ network, augmented with an enhanced ASPP module, first trains on the semantic segmentation dataset to derive image feature representations. Subsequently, a random forest model within the dry matter prediction framework undergoes further training for the purpose of forecasting dry matter content. This comprehensive approach enables the model to establish predictive prowess by internalizing relationships between sample features and the corresponding dry matter content. This orchestrated process leverages input images, feature extraction, and dry matter content prognostication, culminating in a precise prediction of clover's dry matter content. By orchestrating the complete dry matter prediction model, the innate strengths of the DeepLabv3+ network within semantic segmentation tasks are astutely harnessed. This transpositional learning strategy not only truncates training duration but also augments the model's performance in the domain of dry matter prediction.

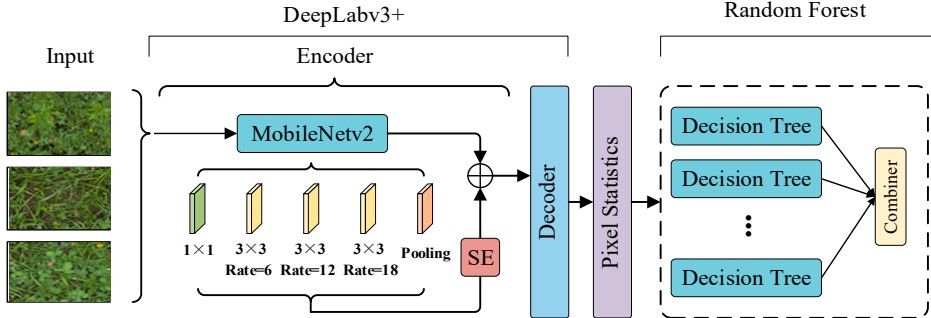

**Figure 1.** A framework for clover dry matter prediction based on semantic segmentation network and RF, where the semantic segmentation network is DeepLabv3+.

*2.2. DeepLabv3+*

Deeplabv3+ [21] represents a semantic segmentation network characterized by its coder–decoder architecture enhanced with a spatial pyramid pooling module, as illustrated in Figure 2. The image input backbone network yields features that serve two crucial roles. Firstly, these features are directly fed into the decoder component, assuming the guise of low-level semantic features. This preserves original image details and local information. Secondly, the ASPP employs dilated convolution with diverse expansion rates to process these features. This operation begets semantic features of varying scales within the image, constituting the high-level decoder features. Dilated convolution alleviates the issue of insufficiently reconstructing information from diminutive objects during the enlargement of the receptive field. Concurrently, the encoder–decoder architecture methodically recovers spatial information, enhancing the discernment of object boundaries. This overarching structure progressively reinstates the spatial characteristics of the image, thereby enabling a more accurate capture of object boundaries. The encoder extracts high-level abstract features from the input image, subsequently conveyed to the decoder. The decoder then employs these features to iteratively delineate more distinct object boundaries through the gradual restoration of spatial information. This hierarchical approach contributes to refined image reconstruction, facilitating the network in faithfully capturing and reinstating object contours from the original image while preserving a wealth of intricate details.

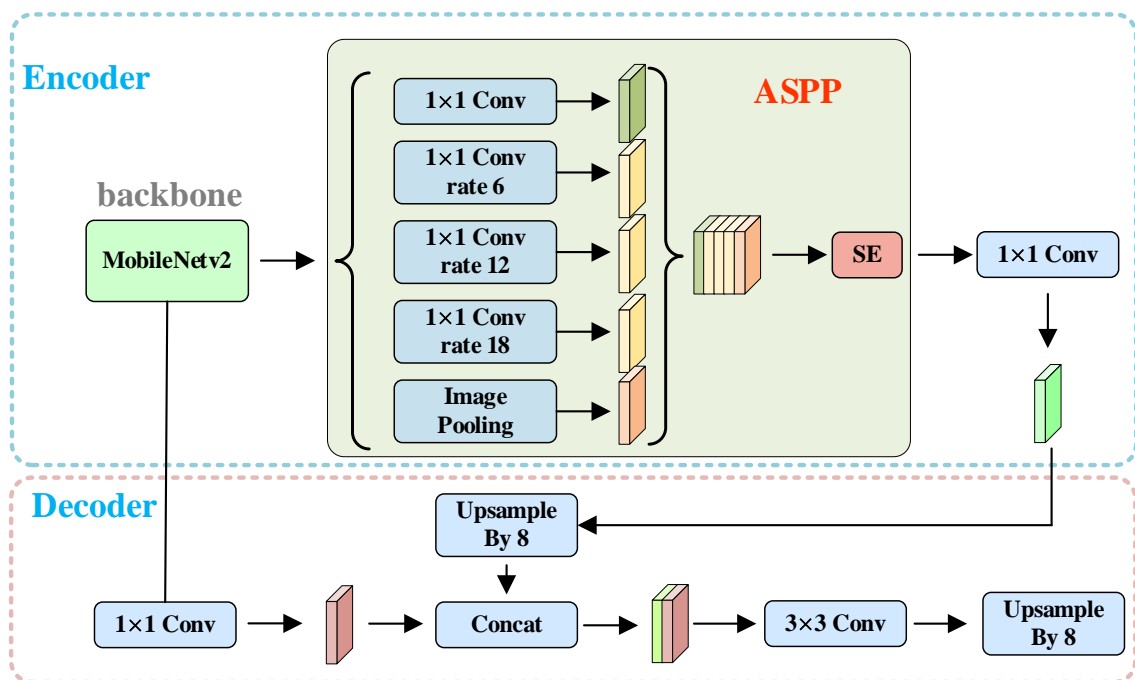

**Figure 2.** Deeplabv3+ network architecture where backbone is MobileNetV2.

2.2.1. MobileNetv2

MobileNetV2 [28] is used here as a backbone. v2, along with v1 [29], is a deep neural network model developed by Google for use in mobile devices and embedded systems. As depicted in Figure 3, the MobileNetV2 authors propose a design strategy called "Inverted Residuals with Linear Bottlenecks" that balances performance and efficiency by introducing deeply separable convolutions and linear bottleneck structures. Deeply separable convolution, a technique that dissects the conventional convolution into two sequential phases—deep convolution and pointwise convolution—achieves a twofold reduction in computational load and parameter count. The deeply separable convolution strategically disentangles the conventional convolution procedure into a sequential process encompassing deep convolution and pointwise convolution. This strategic division culminates in a dual benefit, marked by a pronounced mitigation in computational complexity and

a concomitant reduction in parameter proliferation. Meanwhile, the introduction of linear bottleneck structures can better utilize the network capacity and introduce nonlinear activation functions between each bottleneck layer to improve the feature expression.

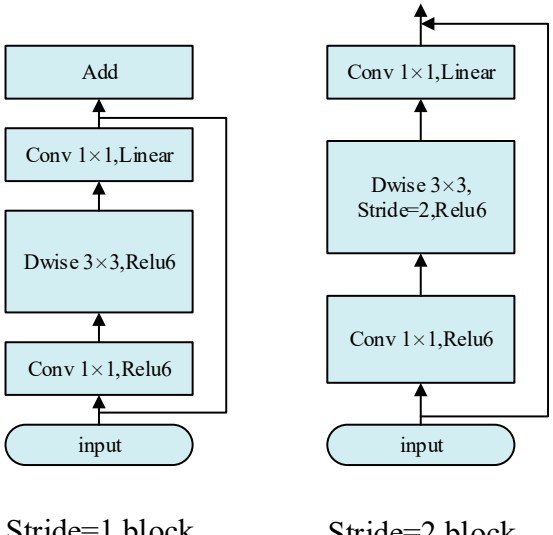

**Figure 3.** Inverse residuals of linear bottlenecks with stride = 1 and stride = 2.

"Inverted Residuals with Linear Bottlenecks" is a key component in MobileNetV2, which is used to introduce non-linear transformations in the network while reducing the dimensionality of the feature mapping. Inverted residuals are an idea opposite to the traditional residual blocks. In a traditional residual blocks, a dimensionality reduction operation (e.g., convolution) is first performed, then a nonlinear transformation is performed, and finally, a dimensionality enhancement operation is performed. Whereas in inverted residual blocks, nonlinear transformations are performed first, followed by dimensionality reduction and dimensionality enhancement operations. This order change can effectively introduce nonlinearities and improve the expressiveness of the model. For linear bottlenecks, in the traditional ResNet structure, the bottleneck layer usually consists of a $1 \times 1$ convolutional layer for dimensionality reduction, a nonlinear activation function, a $3 \times 3$ convolutional layer, another nonlinear activation function and a 1x1 convolutional layer. However, in MobileNetV2, the bottleneck layer is modified to be a linear activation function, which reduces the number of nonlinear transformations and contributes to computational efficiency. "Inverted Residuals with Linear Bottlenecks" refers to the use of inverted residual blocks to introduce nonlinear transformations and a bottleneck layer with a linear activation function in it to reduce the dimensionality of the feature mapping in the MobileNetV2 architecture. This combination allows the network to have a smaller computational and memory overhead while maintaining a higher accuracy for resource-limited scenarios.

Table 1 outlines the network architecture of MoblieNetv2 as adopted in this study. The parameter t represents the expansion factor employed to regulate the augmentation of the input channel count within the bottleneck block. c signifies the number of output channels, signifying the final count of feature channels emerging from the bottleneck block. n indicates the number of repeated blocks, indicating the frequency at which a given bottleneck block is iterated in the network. s, denoting stride, specifies the convolution operation's step size. In contrast to the base model, we have augmented the input image dimensions to amass a more extensive array of features.

**Table 1.** Moblienetv2 network architecture.

| Input | Operator | t | c | n | s |
|---|---|---|---|---|---|
| $1024^2 \times 3$ | conv2d | - | 32 | 1 | 2 |
| $512^2 \times 32$ | bottleneck | 1 | 16 | 1 | 1 |
| $512^2 \times 16$ | bottleneck | 6 | 24 | 2 | 2 |
| $256^2 \times 24$ | bottleneck | 6 | 32 | 3 | 2 |
| $128^2 \times 32$ | bottleneck | 6 | 64 | 4 | 2 |
| $64^2 \times 64$ | bottleneck | 6 | 96 | 3 | 1 |
| $64^2 \times 96$ | bottleneck | 6 | 160 | 3 | 2 |
| $32^2 \times 160$ | bottleneck | 6 | 320 | 1 | 1 |
| $32^2 \times 320$ | conv2d $1 \times 1$ | - | 1280 | 1 | 1 |
| $32^2 \times 1280$ | avgpool $32 \times 32$ | - | - | 1 | - |
| $1 \times 1 \times 1280$ | conv2d $1 \times 1$ | - | k | - | |

### 2.2.2. ASPP Based on Squeeze and Extraction Networks

The conventional convolution employs a fixed-size kernel, calculating each image position with a consistent stride. The receptive field size is contingent upon the kernel dimensions, expanding in line with network layer augmentation. However, this expansion escalates computational demands. In the context of ASPP, dilated convolution introduces the notion of an expansion rate into conventional convolution, governing the kernel's sampling interval through the expansion rate, as illustrated in Figure 4. This innovation enhances the kernel's voids, thus amplifying the receptive field, and consequently extracting more extensive features. In essence, dilated convolution in ASPP employs the concept of expansion rate within conventional convolution, manipulating the sampling stride of the convolutional kernel to augment voids within the kernel. This approach expands the receptive field, yielding features endowed with more expansive receptive fields.

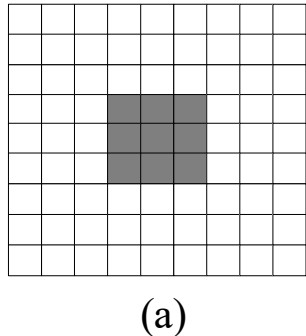
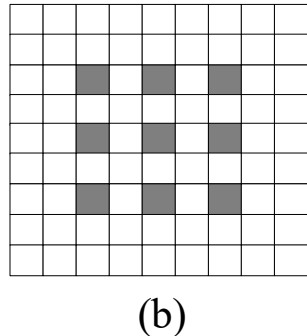

(a)        (b)

**Figure 4.** Dilated convolution schematic with $3 \times 3$ convolution kernel (**a**) Expansion rate of 1, ordinary convolution (**b**) Dilated convolution with expansion rate of 2.

Designating the initial feature map as $X$ and the resultant feature map from the cavity convolution as $Y$, the operation of Dilated convolution can be formalized as follows:

$$Y[i,j] = \sum_m \sum_n X[i + m \cdot d, j + n \cdot d] \cdot K[m,n] \tag{1}$$

Here, $i$ and $j$ represent the spatial coordinates of the resulting feature map, while $m$ and $n$ indicate the coordinates of the convolution kernel. The symbol $k$ pertains to the weight assigned to the convolution kernel at the corresponding spatial position, and $d$ signifies the expansion rate. In contrast, the conventional convolution can be conceived as a specific instance of dilated convolution, characterized by an expansion rate of 1.

Figure 2 shows the model structure of ASPP in DeepLabv3+, which uses multiple parallel convolutional branches, each using a different size of dilated convolutional kernel. In this way, the model can capture features under different sensory fields, which enables the model to better understand the objects in the image as well as their contextual information.

In order to better fuse features at different scales, this paper introduces the SE [30] attention mechanism before the fusion of features at each scale, which can amplify the response to important features and suppress the response to unimportant features through the learning of channel weights, and is able to adaptively learn the correlation between different channels in the input feature map, and then re-calibrate the features of each channel, so as to improve the ASPP in the performance in feature extraction capability.

### 2.3. Dry Matter Prediction

RF represents a potent machine learning methodology that is widely used in data analysis and prediction tasks. It is an integrated learning algorithm that makes predictions by constructing multiple decision trees and synthesizing their results.

The basic principle of random forest is to achieve prediction by combining multiple decision trees. Each decision tree is a classification or regression model that classifies data into different categories or makes predictions of continuous values by progressively dividing the input features. Each decision tree in a random forest is constructed independently, by randomly selecting a subset of features and samples. This randomness makes each decision tree different, thus increasing the diversity of the model. In random forests, the random selection of feature subsets is achieved by randomly selecting a fraction of the features from all the features. This is carried out to reduce the correlation between features and to ensure that each decision tree is able to take into account different feature information. In terms of samples, random forest constructs the training data required for each decision tree by randomly selecting samples with put-back. Such sampling allows the model to have a certain degree of randomness while maintaining the characteristics of the data distribution, improving the generalization ability of the model.

The configuration of the random forest model is depicted in Figure 5. In this case, the input data are the number of pixel values for each classification segmented by the DeepLabv3+ network, and the pixel values are used as input features, which are regressed by constructing different decision trees so that each one of them is regressed, and each one of them adopts a similar structure, but the diversity is guaranteed by the introduction of randomness in the training process. To construct each decision tree, we randomly select a different subset of samples and features and learn the model through a regression algorithm that maps the number of pixel values to the prediction of dry matter. Thus, each decision tree becomes adept at discerning the intricate and nonlinear correlation existing between the array of input pixel values and the outcomes related to dry matter content. After training all the decision trees, we use a combiner to summarize the prediction results of all the decision trees. The combiner uses a simple averaging strategy to determine the final dry matter prediction result.

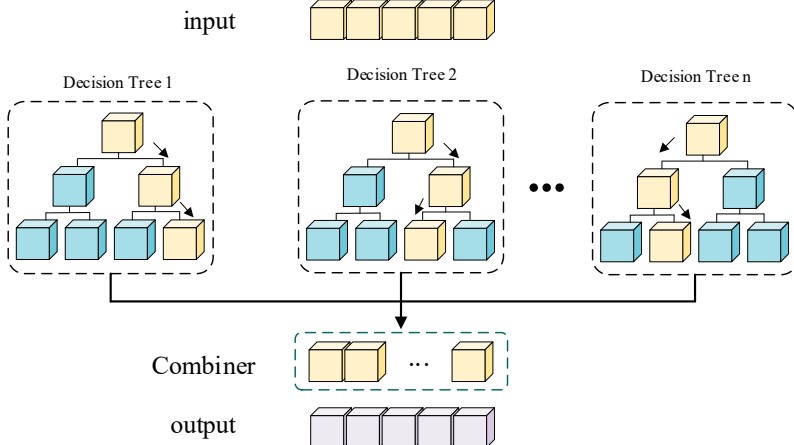

**Figure 5.** Random forest architecture.

*2.4. Experimental Workflow*

The framework illustrated in Figure 1 amalgamates the strengths inherent in both deep learning and machine learning paradigms, culminating in the accurate prognosis of clover dry matter content. The DeepLabv3+ network, revered as a state-of-the-art model extensively harnessed for tasks entailing semantic segmentation, excels in its capacity to discern semantic information within images. Leveraging its hollow-space convolutional pooling pyramid structure, the network adeptly captures image features across varying scales, thereby exhibiting remarkable proficiency in semantic segmentation undertakings. Through the stratification of image features into high-level and low-level components within the encoder module, this methodology astutely exploits multi-scale features, amplifying its acumen in extracting semantic information from images. This strategic framework not only lays a robust groundwork for the subsequent prediction of dry matter content but also underscores the dynamic synergy between image analysis and predictive modeling.

Crucially, the program explores how the features extracted by the DeepLabv3+ network in semantic segmentation can be transformed into useful information for dry matter prediction. In this process, firstly by training on a semantic segmentation dataset, the DeepLabv3+ network learns the semantic distribution of different regions in an image and thus extracts rich image features. These features not only contain information about the appearance of the object, but also capture the relationship between the object and its surroundings.

Subsequently, using the extracted features, this paper designs a dry matter prediction model. The key part of this model is the random forest model, which is widely used in the machine learning field for regression and classification tasks. By learning the relationship between features and dry matter content on a training dataset, the random forest model is able to build a mapping from the features of an input image to the corresponding dry matter content. This predictive model building process is essentially a learning process that allows the model to generalize to unseen images and accurately predict the dry matter content of clover.

Here, the framework of the dry matter predictor based on DeepLabv3+ and random forests can be summarized as Algorithm 1.

---

**Algorithm 1**: Dry matter predictor based on DeepLabv3+ and random forests

---

**Input:**
　　The target sample at the clover image $X[x,y]$;
**Output:**
　　Predicted dry matter of clover M;
　　Segmented image of clover;
**for** each $t$ ranging from 1 to the last clover image $N$
　　1. Image preprocessing, standardization, resize;
　　2. Implementing encoder feature extraction on the basis of the upper part of Figure 2;
　　**3. Introduction of feature extraction network for SE**
　　4. Implementing a featured decoder based on the bottom half of Figure 2;
　　5. Implementing pixel-level segmentation of clover images, counting the number of pixels;
　　6. Number of pixels combined with real dry matter to construct random forest models;
　　**7. Constructing a Random Forest Regression Model**
　　8. Construct a random forest regression based on Figure 5;
　　9. Predict dry matter in target image;
**end for**

---

## 3. Experiments

In the task of dry matter prediction, there is a need to go for evaluating the pixel segmentation performance of DeepLabv3+ as well as the accuracy of dry matter content prediction, so the results are evaluated from these two aspects. Our experiments are conducted in two aspects: the influence of distinct backbone networks on the efficacy of

semantic segmentation, and the performance evaluation of the random forest model n dry matter content prediction.

### 3.1. GrassClover Image Dataset

The GrassClover dataset [31] is a comprehensive resource designed for semantic segmentation of high-resolution images captured in outdoor agricultural environments. This dataset focuses on densely vegetated categories within the grass–clover domain, comprising 8000 finely annotated synthetic high-resolution images, each with pixel-level perfect annotations. Additionally, it includes 15 pixel-level annotated images with a resolution of 1 million pixels and 31,600 unlabeled images collected from five different locations using three different capture platforms: NiKon d810a, Sony a7 mk1 and IDS 3280CP. Among them, 435 images are annotated with the biomass composition present in the images.

The main image categories in the dataset include perennial grass (Lolium perenne), red clover (Trifolium pratense), white clover (Trifolium repens), soil and weeds. In this context, the "weeds" category encompasses various species, predominantly dandelions, thistles, pasture grass, yellow rattle and creeping buttercup.

For our research, we focus on a subset of the dataset, utilizing 8000 finely annotated high-resolution images and 435 images annotated for biomass content. The synthetic nature of these images allows controlled experiments, including detailed pixel-level segmentation and instance annotations, facilitating effective supervised learning. The collection locations of these images cover two experimental sites and real-world scenarios on three dairy farms in Denmark. These samples were collected between May and October of 2017 and May to October of 2018, ensuring a comprehensive representation of seasonal variations and growth cycles.

#### 3.1.1. Semantic Segmentation Dataset

In semantic segmentation experiments, we leveraged a set of 8000 synthetically generated images from the GrassClover dataset, as depicted in Figure 1. The primary aim was to execute precise semantic segmentation targeting specific plant species. To refine the experiment's clarity, we reorganized the initially diverse image labels based on Figure 6. The modified labels are visualized in Figure 7. Subsequently, the image labels underwent reclassification and were grouped into five main categories: red clover, white clover, soil, grass and weeds. These categories were then further divided into both training and test sets at a ratio of 9:1, ensuring a balanced distribution of data for effective model training and evaluation. It is noteworthy that the labels of these synthetic images were meticulously annotated, underscoring the high quality and accuracy of the annotations, thereby providing a reliable foundation for the experiments.

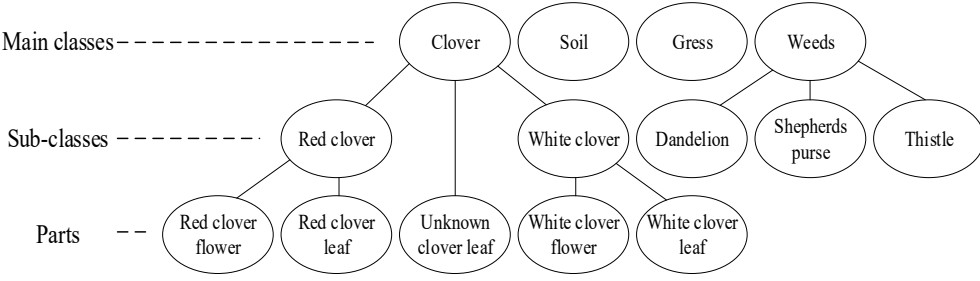

**Figure 6.** Labels hierarchy of the synthetic images, with the label annotation positions at the lowest level of the hierarchy [31].

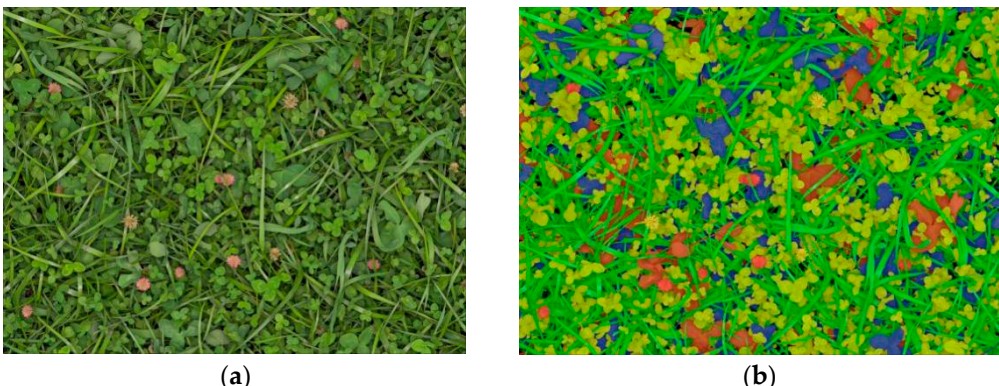

(**a**)                                        (**b**)

**Figure 7.** (**a**) Synthetic training image (**b**) Reclassified image semantic labels for major categories, where soil is black, grass is green, white clover is yellow, red clover is red, and weeds are blue.

### 3.1.2. Dry Matter Prediction Dataset

In this study, the dataset used for biomass prediction experiments is also derived from GrassClover. This dataset comprises 435 authentic images, each meticulously annotated with biomass information. These genuine images were obtained through on-site biomass sampling in agricultural fields, with precise annotations reflecting the accurate biomass content of the vegetation in each image. This dataset serves as a valuable reference for our biomass prediction experiments, providing real-world context.

Each biomass sample includes the following components:

1. A canopy image of a defined 0.5 m by 0.5 m of grass clover preceding the cut.
2. A composition of the harvested biomass with stems located in the square.

After cutting the plants at a height of 5 cm, all plant samples were separated into rye-grass, clover and weeds. Among them, 272 samples underwent further subclass separation into red clover and white clover. After drying the samples, each fraction was individually weighed to determine the dry matter yield and composition. A total of 435 biomass samples were collected. The sampling spanned the seasons of 2017, with additional samples in 2018.

By utilizing this dataset of real images annotated with biomass information from GrassClover, we aimed to further validate the robustness and accuracy of our biomass prediction model. These images from real-world scenarios will provide training and evaluation data that closely align with practical applications, thereby enhancing our confidence and understanding of the performance of the biomass prediction model. We used 261 dataset images with biomass labels from this dataset.

### 3.2. Experimental Environment

The experimental platform in this paper is the Window11 operating system, the CPU is Intel(R) CORETM i9-10900k CPU@3.7GHZ, the computational graphics card for deep learning is dual GeForce RTX 4090, corresponding to a graphics memory size of 48 GB, and the version of the deep learning framework torch was used.

### 3.3. Experimental Results of DeepLabv3+ Network

In order to verify the feature extraction performance of the feature extraction network used in this paper, it is evaluated according to the general evaluation criteria of semantic segmentation, and for the evaluation of DeepLabv3+ network we use *Precision*, *IoU*, and *PA*(Pixel Accuracy), the formula for the particular index is outlined as follows:

$$Precision = \frac{TP}{TP + FP} \tag{2}$$

$$IoU = \frac{TP}{TP + FP + FN} \tag{3}$$

$$PA = \frac{TP + TN}{TP + TN + FP + FN} \tag{4}$$

where *TP* corresponds to instances where a sample was predicted as belonging to a positive category while its true label also indicated a positive category; *FN* corresponds to cases where a sample was erroneously predicted as a counterexample, but its true label indicated a positive example; *FP* pertains to situations where a sample was inaccurately predicted as a positive example, while the true label suggested a counterexample; and *TN* represents scenarios where a sample was correctly identified as a counterexample, aligning with its true label.

To ascertain the aggregate performance of each evaluation metric, the cumulative sum of the evaluation metrics across all categories was divided by the total number of categories. The training results for DeepLabv3+ are depicted in Figure 8.

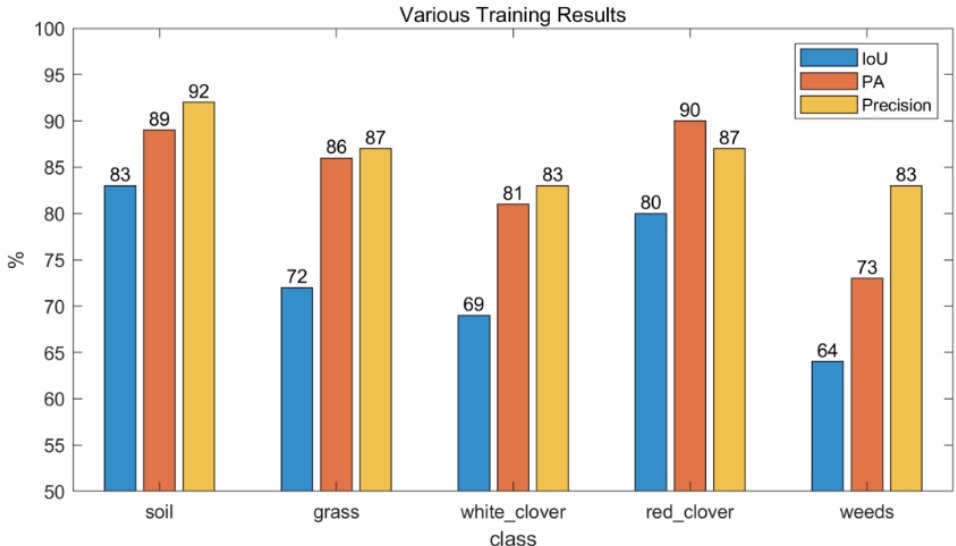

**Figure 8.** Training results of DeepLabv3.

To assess the performance of the method proposed in this paper, the results of our experiments on GrassClover are presented in the form of bar charts showing the IoU, PA and precision metrics for each category. In Figure 6, we can clearly observe that our methodology excels in terms of recognition accuracy across diverse vegetation categories. Our method exhibits relatively high IoU, PA and precision scores, which indicates that our approach demonstrates the capability to accurately identify and segment pixels within the category of clover. The overall recognition results for weeds are poorer compared to the other categories; this is because the weeds category merges the three weeds, as well as the five categories of unknown_clover_leaf and unknown_clover_flower when merged, which can lead to a slightly poorer recognition due to the complexity of the texture, shape or appearance features. However, overall in terms of clover category recognition, our method performed well and provided a reliable basis for further clover dry matter prediction. Particularly in terms of the level degree of mIoU, the model we used performs 18.5% better on the GressClover dataset than on the alignment proposed by Skovsen S [31].

To assess the feature extraction prowess of the devised DeepLabv3+ network, we compared the DeepLabV3+ models of several different backbone networks and the modeling results are shown in Table 2. The magnified local details of various backbone networks are presented in Figure 9.

**Table 2.** Performance comparison of DeepLabv3+ across different backbone networks.

|  | Size | mPrecision | mIoU | mPA |
|---|---|---|---|---|
| MobileNetV2 | 22.4 MB | 83.69% | 71.15% | 82.20% |
| Xception | 209 MB | 81.82% | 66.93% | 78.48% |
| MobileNetV3 | 15.7 MB | 79.71% | 64.92% | 77.81% |
| MobileNetV2 + SE | 23.2 MB | 85.37% | 73.50% | 83.85% |

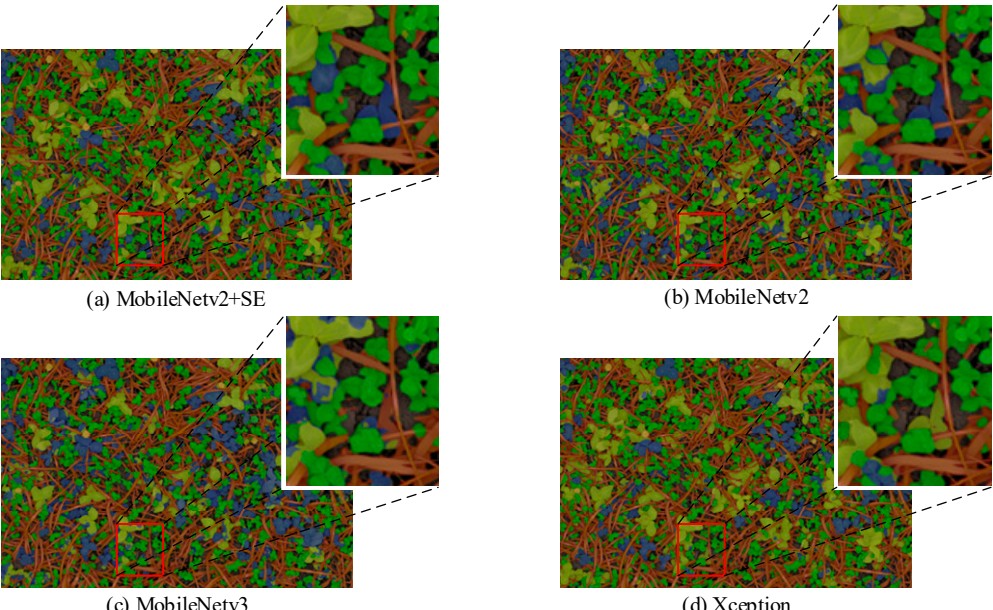

(a) MobileNetv2+SE

(b) MobileNetv2

(c) MobileNetv3

(d) Xception

**Figure 9.** Local pixel enlargement of each model. DeepLabV3+, built by MobileNetv2+SE, has a clearer recognition boundary and better recognition accuracy in terms of segmentation performance.

*3.4. Experimental Results of Dry Matter Prediction*

The model trained in the DeepLabv3+ network was used on the dry matter prediction dataset to perform pixel-level segmentation of clover images to obtain semantic information and count the pixels in the image for dry matter information prediction. For the regression fitting of random forest, we carried out two scenarios here: Scheme 1: use all the data in the dataset to construct the random forest fitting model; Scheme 2: in order to show the feasibility of this model in practical applications and to test its generalization performance, the dataset was partitioned into distinct training and test sets, distinguished by an 8-2 ratio, for training as well as validation of the model.

According to Figure 1, we constructed a complete dry matter prediction model using the DeepLabv3+ network to output the pixel values of the categories, and processed these statistics by constructing a random forest regression model to further mine the association and contextual information between the categories, as well as between the categories and the dry matter content.

We used the root mean square error (RMSE), and the mean absolute error (MAE) for the assessment of the predicted values. It is calculated as where y is the true dry matter content value and is the predicted value predicted by the model.

$$RMSE = \sqrt{\frac{1}{N}\sum_{t=1}^{N}(y - \hat{y})^2} \tag{5}$$

$$MAE = \frac{1}{N}\sum_{t=1}^{N}|y - \hat{y}| \tag{6}$$

The random forest linear fitting of the complete dataset has a very high fitting performance, with a considerable improvement in the RMSE as well as the MAE in all categories compared to the collinear first-order linear model designed by Skovsen S et al. The random forest model is able to capture nonlinear relationships by integrating the results of multiple decision trees, and is able to better model complex nonlinear functions, with good performance capabilities in mapping from image pixel values to dry matter content. The training results are shown in Table 3. The comparison of the predicted dry matter scores with the true values obtained is shown in Figure 10.

**Table 3.** Scheme 1: random forest fitting performance for the complete dataset.

| Method | | Grass | White Clover | Red Clover | Clover | Weeds |
|---|---|---|---|---|---|---|
| RF | RMSE [%] | 5.15 | 3.25 | 2.80 | 4.71 | 1.44 |
| | MAE [%] | 3.57 | 2.37 | 1.54 | 3.37 | 0.83 |
| First order | RMSE [%] | 9.05 | 9.91 | 6.50 | 9.51 | 6.68 |
| linear [31] | MAE [%] | 6.85 | 7.82 | 4.65 | 7.62 | 4.87 |

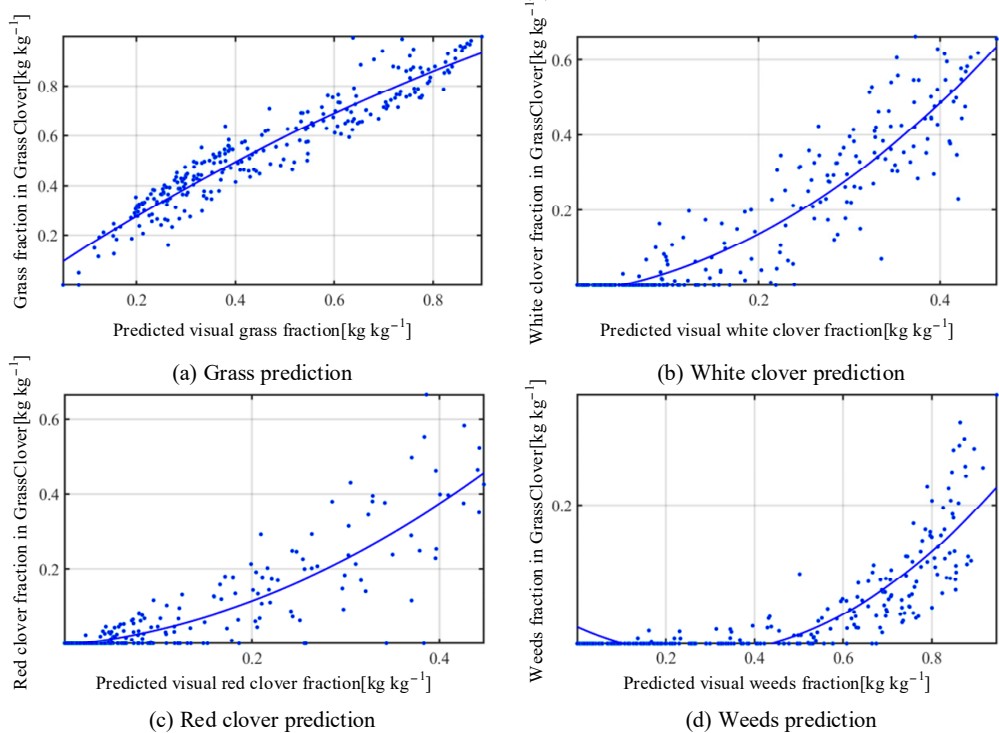

**Figure 10.** Scheme 1 Plot of the points of fit between the true dry matter mass fraction and the predicted mass fraction, and its second-order fit curve.

In Scheme 2, 80% of the 261 samples are selected as the training set, and 20% of the data are used as the test set, and the random forest regression model for dry matter prediction is reconstructed, and the performance of the test set is shown in Table 4, which shows that, relative to the collinear line, there is a certain degree of improvement in the performance, except for the grass class.

**Table 4.** Scheme 2: Training performance of dry matter prediction on random forest models.

| | Grass | White Clover | Red Clover | Clover | Weeds |
|---|---|---|---|---|---|
| RMSE [%] | 16.26 | 4.05 | 3.92 | 2.94 | 3.47 |
| MAE [%] | 10.76 | 3.00 | 2.44 | 2.11 | 2.58 |

## 4. Conclusions

In this study, we propose an innovative approach to the dry matter prediction problem in clover production sites, combining the DeepLabV3+ semantic segmentation model in deep learning and the random forest regression model in machine learning. Based on the in-depth exploration and improvement of the DeepLabV3+ model, we pay special attention to its ASPP structure and introduce the SE attention network, which strengthens the network's ability to characterize the features at the feature channel level. This fusion is important for the fine-grained prediction of dry matter content of different crops in semantic segmentation tasks.

In order to construct a dry matter prediction model, pixel-level semantic segmentation of the dry matter prediction dataset was performed using the adapted DeepLabV3+ network, resulting in finely segmented clover images. The information about the dry matter content of the crop was obtained by counting the number of these images at the pixel level. In order to predict dry matter content more accurately, a random forest regression model was further introduced to model the relationship between features such as pixel-level counts and actual dry matter content, thus achieving the reliable prediction of dry matter content.

The contribution of this study is to enhance the model's ability in capturing crop features by optimizing the structure of the DeepLabV3+ model and incorporating the SE attention mechanism. Meanwhile, a comprehensive dry matter prediction framework is constructed by combining pixel-level semantic segmentation with random forest regression, providing a novel and effective approach for dry matter management and prediction in clover production. This has significant potential for optimizing agricultural production and achieving precision agricultural management, and also demonstrates a useful example for the integration of deep learning and machine learning in interdisciplinary research.

Furthermore, it is imperative to consider the practical implementation of our proposed method in real-world agricultural settings, where end users may not possess advanced computer skills. To address this, we envision a user interface that simplifies interaction with our system. This conceptual interface would include intuitive controls, visual aids and easy-to-understand prompts. Additionally, it should offer flexibility to accommodate different skill levels among agricultural practitioners. While the intricacies of the interface design are beyond the scope of this paper, we emphasize the significance of creating a user-centric interface that enhances the usability of our method in the agricultural domain. Such an interface would contribute to smoother integration into existing agricultural workflows, ultimately maximizing the accessibility and utility of our approach for the target user base.

**Author Contributions:** Conceptualization, J.F., Y.Z. and Y.J.; methodology, Y.J.; software, Y.J.; validation, J.F., Y.J. and Y.Z.; formal analysis, Y.J.; investigation, Y.J.; writing—original draft preparation, Y.J.; writing—review and editing, Y.J.; visualization, Y.J.; supervision, J.F. and Y.Z.; project administration, J.F. and Y.Z.; funding acquisition, J.F. All authors have read and agreed to the published version of the manuscript.

**Funding:** This work was supported by the Inner Mongolia Scientific and Technological Project under Grant (2023YFJM0002, 2022YFSJ0034).

**Data Availability Statement:** The GrassClover image dataset is made publicly available at https://vision.eng.au.dk/grass-clover-dataset, accessed on 26 October 2023.

**Conflicts of Interest:** The authors declare no conflict of interest.

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
