# Peer review of "Clover Dry Matter Predictor Based on Semantic Segmentation Network and Random Forest"

_applsci, doi:10.3390/app132111742_

Round 1

Reviewer 1 Report

Comments and Suggestions for Authors

Dear Authors

I have had the pleasure of reviewing the manuscript titled "Clover dry matter predictor based on semantic segmentation network and Random Forest". I would like to commend the authors for their significant contribution to the field and recommend this paper for publication in Applied Sciences. Here are some of the notable strengths of the paper:

- The authors have undertaken extensive experiments, applying their trained model to a dry matter prediction dataset. The results, showcasing an improvement in the performance of semantic segmentation, are commendable and point towards the robustness of the proposed method.
- The paper bridges the gap between advanced machine learning techniques and agricultural applications, highlighting the growing significance of AI in diverse fields.

- The manuscript is well-organized, making it accessible for both experts in the field and readers who might be new to the topic. The logical flow of content aids in understanding the complex methodologies employed.

In conclusion, the manuscript is a valuable addition to the literature, offering novel insights and methodologies that could pave the way for future research in the domain. I believe the paper is well-suited for publication and will be of interest to its readership.

Author Response

Dear Reviewer,

I would like to extend my sincere gratitude for your invaluable feedback on our manuscript titled "Clover dry matter predictor based on semantic segmentation network and Random Forest." Your expert insights and encouragement are greatly appreciated and hold significant importance for our research work.

First and foremost, I wish to express my gratitude for your positive recognition of our research. Your acknowledgment of the extensive experiments we conducted on the dry matter prediction dataset, as well as the validation of the reliability of our proposed method through improved semantic segmentation performance, is truly motivating. Your support encourages us to continue our in-depth exploration and continuous improvement of our approach.

Your observation that our paper successfully bridges advanced machine learning techniques with agricultural applications, emphasizing the importance of artificial intelligence across diverse fields, aligns perfectly with the core objective of our research. We are genuinely grateful for your recognition of this aspect, and we hope that our work can contribute further innovation and practicality to the field of agriculture.

Lastly, your appreciation of the structure and presentation of our paper brings us great satisfaction. We have always aimed to ensure that our manuscript caters to both experts in the field and newcomers, providing a clear understanding of our complex methodologies. Your acknowledgment reassures us that we have succeeded in this regard.

Once again, thank you for your diligent review and valuable suggestions. Your support is of utmost significance to our research, and we will earnestly consider your feedback to enhance our work. If you have any questions or require additional information, please do not hesitate to reach out.

With heartfelt appreciation,

Yin Ji

Reviewer 2 Report

Comments and Suggestions for Authors

Overall, this is a very interesting and well-written paper. It considers images of clover and applying semantic segmentation makes a prediction regarding the content of dry matter content in the clover. Since the clover is one of the main animal feeds for domestic animals, the work is of practical importance.

The paper is well-written, easy to read and very suitable for the audience of Applied Sciences. 

The paper starts with an introduction, containing a survey of the literature and highlighting the paper contributions. It is followed by a section detailing the methods. It describes the overall architecture, the semantic segmentation network, the random forest used for prediction of the clover dry matter, and the respective algorithm. Several illustrations are used to better convey the ideas.

The third section is devoted to the experiments, which prove the usefulness of the proposed approach.

Finally, the section Conclusions discusses the results and the prospective uses in agriculture.

The literature contains 31 sources, 14 of which are from the last 5 years.

Overall, the paper merits publication and is of clear practical importance.

I have a few suggestions for improvement:

1) The method is intended to be used in the agricultural practice, where the employees may not necessarily be proficient in working with computer systems. It may be useful to describe the prospective interface of the system, at least at conceptual level. 

2) The authors may consider including at the end of the introduction a paragraph describing each section with one sentence.

3) Some stylistic editing of the language may be useful. For example, the second sentence of the abstract "The objectives of this study is to propose a method for predicting the dry matter content of clover based on semantic segmentation network, which is realized by constructing a prediction model based on the visible image information to predict and analyze the dry matter content in clover. " contains three related words - predicting, prediction, and predict. This should be improved using rewording.

Comments on the Quality of English Language

Overall, the English is good. Some stylistic editing can be used to improvement the text - see my example above.

Author Response

Dear Reviewer,

I would like to sincerely thank you for the thoughtful and constructive feedback you provided on our paper, "Clover dry matter predictor based on semantic segmentation network and Random Forest". Your insights and suggestions were crucial in enhancing the quality and clarity of our paper, and we greatly appreciate the time and effort you put into the review process.

Your assessment that you found our paper interesting and well-written is encouraging, and we are glad that you recognized the practical importance of our research in the context of agricultural applications, especially in the context of clover as an important component of animal feed.

We also appreciate your comments on the structure of our paper, including the introduction, methods, experiments and conclusion sections. Your feedback reaffirms our efforts to present a well-organized and accessible paper for the readers of Applied Science.

  • In light of the valuable feedback from the reviewer, it is essential to address the practicality of implementing our proposed method in real-world agricultural settings. As highlighted by the reviewer, it is crucial to consider the end users, who may not necessarily possess advanced computer system proficiency. To address this concern, we acknowledge the importance of describing the prospective system interface, even if at a conceptual level. This step can significantly enhance the usability and adoption of our approach in agricultural practice, making it more accessible to a broader audience.We added this paragraph to the conclusion.
  • Thank you for providing suggestions regarding the introduction section. We greatly appreciate your input, as it will help improve the clarity and comprehensibility of the paper. In accordance with your advice, we will add a paragraph at the end of the introduction to briefly describe each section in one sentence, which will assist readers in better understanding the structure and content of our paper. This will contribute to ensuring that our paper is more easily readable and understandable.
  • Thank you for mentioning the issue of repetitive use of the word 'prediction' in the second sentence of the abstract. It is indeed worth addressing. We will work on improving that sentence to reduce redundancy and enhance clarity of expression. We will ensure that the writing style throughout the article becomes more fluid, making it easier for readers to understand our research.

Thank you once again for your thorough review and your efforts to ensure the quality of our paper. Your comments have played a huge role in enhancing our paper and we appreciate the contribution of your expertise and guidance in the review process.

If you have any further comments or suggestions, please feel free to share them. Your feedback is highly valued and will continue to provide significant assistance in our research endeavors.

Sincerely thanking you for your support.

Yin Ji

Reviewer 3 Report

Comments and Suggestions for Authors

The article presents a deep learning applications (via semantic segmentation network and rgb images) for identification and biomass estimation of clover. In general the deep learning aspect is well-explained. However, the dataset preparation method is not detailed. Since this study presents an application on clover identification and biomass estimation, the dataset preparation details are important to ensure that the ground truth is valid. For example, how are the labels verified for clover identification as there could be variation between labelers, particularly for dense and complex images such as the data presented in this study which has multiple objects within one image, and how are the biomass measured experimentally? Are the clover data collected from clovers within the same maturity?

The quality of the paper will be better with this information. Thank you.

Comments on the Quality of English Language

I could not find obvious language usage error in this paper. 

Author Response

Dear Reviewer,

I sincerely appreciate your invaluable feedback on my paper titled "Clover dry matter predictor based on semantic segmentation network and Random Forest." It is truly an honor to continuously improve my research under your guidance. Below are detailed responses to the issues you raised:

  • Elaboration on the Data Set Preparation Method:

Thank you for your attention to the data set preparation method. In my study, I utilized synthetic images as a semantic segmentation data set, ensuring that labels were perfectly annotated. This implies that each image underwent highly consistent and accurate labeling. I will enhance the paper by providing detailed explanations in this regard, emphasizing the high-quality annotation of labels and how uniformity among annotators was ensured.

  • Clover Data Collection and Maturation Discrepancies:

Concerning the collection of clover data and the issue of different maturities, I will explicitly state in the paper that all clover data were collected under various maturity conditions. Specifically, samples were collected from May to October in both 2017 and 2018, ensuring maturity variations during this period. This will aid readers in better understanding the diversity of the experiment and the comprehensiveness of the results.

  • Experimental Measurement of Biomass:

I will further elaborate on the experimental measurement method of biomass to ensure readers understand the feasibility and accuracy of the entire process. Specific details on how biomass was measured and the experiment's intricacies will be provided to enhance the paper's completeness.

Once again, I appreciate the suggestions you provided for my research. Your guidance is crucial for enhancing the quality and credibility of the study. If you have any additional suggestions or require further information, I am eager to listen and will ensure timely and appropriate modifications.

Thank you for your time and professional advice. I look forward to your further review of the revised paper.

Sincerely,

Yin Ji
